# Multisource Fusion UAV Cluster Cooperative Positioning Using Information Geometry

Chengkai Tang [1], Yuyang Wang [1], Lingling Zhang [2,*], Yi Zhang [1] and Houbing Song [3]

[1] School of Electronics and Information, Northwestern Polytechnical University, Xi'an 710072, China
[2] School of Marine Science and Technology, Northwestern Polytechnical University, Xi'an 710072, China
[3] Department of Electrical, Computer, Software, and Systems Engineering,
Embry-Riddle Aeronautical University, Daytona Beach, FL 32114, USA
[*] Correspondence: llzhang@nwpu.edu.cn

**Abstract:** Due to the functional limitations of a single UAV, UAV clusters have become an important part of smart cities, and the relative positioning between UAVs is the core difficulty in UAV cluster applications. Existing UAVs can be equipped with satellite navigation, radio navigation, and other positioning equipment, but in complex environments, such as urban canyons, various navigation sources cannot achieve full positioning information due to occlusion, interference, and other factors, and existing positioning fusion methods cannot meet the requirements of these environments. Therefore, demand exists for the real-time positioning of UAV clusters. Aiming to solve the above problems, this paper proposes multisource fusion UAV cluster cooperative positioning using information geometry (UCP-IG), which converts various types of navigation source information into information geometric probability models and reduces the impact of accidental errors, and proposes the Kullback–Leibler divergence minimization (KLM) fusion method to achieve rapid fusion on geometric manifolds and creatively solve the problem of difficult fusion caused by different positioning information formats and parameters. The method proposed in this paper is compared with the main synergistic methods, such as LS and neural networks, in an ideal scenario, a mutation error scenario, and a random motion scenario. The simulation results show that by using UAV cluster movement, the method proposed in this paper can effectively suppress mutation errors and achieve fast positioning.

**Keywords:** cooperative positioning; multisource fusion; information geometry; UAV

## 1. Introduction

The UAV has the characteristics of not being affected by terrain, fast speed, small size, etc. Therefore, it has shown great development prospects in the fields of logistics transportation, security monitoring, information collection, and traffic guidance in smart cities [1]. However, due to load limitations and the occlusion effect of urban canyons, a single UAV provides low coverage, which greatly limits its application in smart cities. While a lightweight UAV cluster has attracted more attention because of its wide coverage and high transportation efficiency, the main direction of UAV development is towards applications in cities [2]. However, unlike a single UAV, the positioning between UAVs in a cluster is the core difficulty in the application of UAV clusters. Compared with traditional node positioning, UAV cluster positioning has the characteristics of a three-dimensional position, large dynamics, and easy interference. The main positioning information of a UAV cluster comes from satellite navigation and inertial navigation, supplemented by wireless base station navigation, laser navigation, visual navigation, and other technologies. Due to inherent problems, various types of navigation sources cannot realize the positioning service of a UAV cluster in the whole area and environment. For example, navigation satellites are easily blocked by urban canyons, resulting in the loss of UAV navigation

signals [3]; wireless base stations have limited navigation transmission distance and cannot provide positioning services outside the area [4]; LiDAR is greatly affected by weather and atmosphere; the beam is narrow, and performance drops significantly in cloudy, rainy, and foggy environments [5]; and visual navigation requires a huge amount of image processing in complex and dynamic environments, exhibits poor real-time performance, and has a large impression of the light-receiving environment [6]. Therefore, using the relative position of a UAV cluster to realize the rapid fusion of distributed multisource navigation system information is the main problem of UAV cluster positioning.

The existing distributed node positioning method mainly realizes cooperative information fusion positioning through ranging, direction finding, and information interaction. This fusion method mainly includes two categories. The first category is the fusion of positioning data represented by neural networks [7–9], and the other is the fusion of localization results represented by the Kalman filter [10–12]. Both types of algorithms have problems of high computational complexity and poor real-time performance, and thus they are not suitable for the cooperative positioning of UAV clusters. Therefore, using the various types of navigation sources independently distributed in the UAV cluster to achieve fast and high-stability fusion is the main problem of the coordinated positioning of the UAV cluster.

As the current mainstream positioning information fusion algorithm, the Kalman filter and its extended series of filtering algorithms [13] can not only estimate the time domain information of stationary random processes but can also estimate nonstationary random processes. However, the positioning information is output by different types of navigation sources. The format and frequency of the parameters are completely different, resulting in high computational complexity and a long fusion time, which cannot meet the needs of UAV clusters for performing the rapid fusion of different types of navigation sources. With the rapid development of artificial intelligence, intelligent information fusion by neural networks has also developed rapidly, and these can realize the rapid fusion of various types of navigation information, but they also have the problems of requiring a large amount of data and exhibiting poor real-time performance. To solve the problem of difficulty in fusion positioning caused by factors such as the time-space asynchrony of distributed multitype navigation source information in UAV clusters, the loss of navigation source information, and the rapid movement of UAV clusters, this paper proposes a multisource information geometry-based approach. The main innovation points of the fusion UAV cluster positioning method are as follows:

1.　The information from various navigation sources carried by the UAV cluster is creatively transformed into an information probability model, the time and frequency parameters of various types of navigation information of the UAV cluster are unified, and a simulation scenario is established to verify the model.
2.　A multisource fusion UAV cluster localization method based on information geometry is proposed. The method utilizes the correlation between the information probability of the UAV navigation sources and the positioning accuracy, calculates the accuracy probability function of the navigation source information, establishes the probability geometric manifold of the navigation source information, and fuses multiple probability density functions to obtain the positioning result.
3.　Simulation tests of the proposed UCP-IG model in ideal scenarios, sudden loss of navigation information scenarios, and random motion scenarios are carried out. The test results show that the UCP-IG method proposed in this paper can effectively improve the stability of UAV clusters. In the case of a loss of human–machine navigation information, errors can also be effectively suppressed.

The rest of the paper is organized as follows. In the following section, we present the cooperative-positioning-related work. Section III introduces the system model. Then, the multisource fusion UAV cluster localization method based on information geometry is introduced in Section IV. Section V contains a detailed analysis of the results, and Section VI presents the concluding remarks.

## 2. Related Work

With the application of UAVs in unmanned transportation applications, traffic management, and other basic core projects of smart cities, the relative positioning between UAV clusters has become the core basis for UAV cluster applications [14]. With the improvement of the accuracy of the GNSS system and the enhancement of the navigation signal by low-orbit communication satellite systems, satellite navigation has become the main method of the cluster positioning of UAVs. However, due to the serious occlusion of urban environments, the signals of some UAVs will always be lost during the flight of a UAV cluster. The development of the coordinated navigation and positioning of UAV clusters has become a hot topic for UAV cluster applications.

UAV cooperative positioning mainly evolved from wireless sensor network cooperative positioning. Early UAV cooperative positioning methods mainly achieved cooperative positioning through the LS method [15], MSE method [16], etc. This kind of method has the advantage of fast positioning speed, but it is easily disturbed by environmental errors.

Due to the jittery characteristics of UAVs, reducing the error has become the main problem of UAV positioning. Techniques such as Bayesian estimation [17] and non-Bayesian estimation [18] are used for UAV cooperative positioning, but the disadvantage of Bayesian estimation methods is the large communication and computational overhead required. The research work discussed in [19] proposed a cooperative positioning fusion technique based on a particle filter (PF), but it is difficult to solve the problem of particle degradation and depletion [20]. To this end, [21] and [22] proposed a positioning estimation method based on the extended Kalman filter (EKF) and unscented Kalman filter (UKF), respectively. Non-Bayesian estimation cooperative positioning methods mainly include the least squares (LS) [23] estimation method and the maximum likelihood (ML) [24] method. The research work discussed in [25] uses weight compensation combined with the LS method to achieve UAV cluster positioning, which can reduce the impact of environmental errors. The study proposes a new hybrid cooperative positioning scheme based on distance and angle measurement; that is, a modification of the TOA-AOA-based and AOA-RSS-based linear least squares (LLS) hybrid scheme. Based on the ranging information between the man–machine and base station, an optimized version of LLS estimation was proposed, which further improves the positioning performance but limits the scope of use. Tomic S et al. [26] proposed a new method based on received signal strength (RSS) and convex optimization. By deriving nonconvex estimates, the search problem of the global optimal solution was solved, but it could not meet the high mobility characteristics of UAVs, and the stability of positioning accuracy was poor.

In recent years, neural networks have also been used in the field of cooperative positioning due to their advantages of arbitrary nonlinear mapping of input and output. The research work discussed in [27] used BP neural-network-assisted EKF and UKF to achieve cooperative positioning, and the results showed that using a BP neural network to optimize nonlinear filtering could improve filter estimation performance. The research work discussed in [28] proposes a layered sensor fusion method with an artificial neural network (ANN) to solve the self-localization problem of mobile robots. This method uses octagonal sonar, digital compass, and wireless network signal strength measurement to determine the position of autonomous mobile robots. Multi-layer perceptron (MLP) is used in combination with supervised learning and back-propagation techniques to train the network by layered fusion steps and determine the robot's position on the map. However, this type of method needs to train various parameters and the calculation amount and time are too large, so it cannot meet the real-time requirements of UAV cluster cooperative positioning.

Therefore, how to realize the cooperative positioning of UAV clusters in an unknown, complex, and dynamic environment and how to maximize the real-time performance of the algorithm and reduce the amount of calculation require further research. For this reason, an increasing number of scholars have introduced graph theory knowledge into cooperative positioning. For instance, Ihler A proposed the nonparametric belief propagation algorithm [29]. The basic idea of this algorithm is a method based on a graph model,

which models the localization problem as an inference problem, allowing it to perform a distributed estimation process, but it has problems of high computational complexity and poor real-time performance. Starting from estimation theory and factor graphs [30–32], Wymeersch H et al. proposed a distributed cooperative positioning algorithm, SPAWN (sum-product algorithm over a wireless network), which transformed the node localization problem in cooperative networks into variables. The approximate edge posterior distribution of variable nodes is obtained by updating and transmitting confidence information on the factor graph. However, the algorithm requires a large amount of computation and large communication bandwidth and has poor positioning performance in dynamic scenarios. On the basis of graph theory, information geometry is widely used as an effective tool for solving nonlinear and random problems. Compared with the mathematical theory of the Euclidean framework, information geometry theory can more effectively solve some nonlinear and random problems in the information field. Information geometry methods have made considerable progress in statistical signal processing, parameter estimation, and filtering. Reference [33] refers to the concept of information geometry to radar signal processing and proposes a corresponding radar signal detection method, but how to use the information geometry distance to realize multi-UAV cooperative positioning is currently unaddressed. This paper takes advantage of the unified parameters of the information geometric model, combined with the K–L fusion method, which can effectively improve the positioning stability of the UAV cluster and the ability to suppress mutation errors.

### 3. System Model

Due to the limited transportation weight and quantity of a single UAV, distributed UAV clusters can greatly improve transportation efficiency and reduce costs, but the mutual distance and control between clusters require high-precision, real-time positioning information support. The cooperative positioning network of the UAV cluster studied in this paper is shown in Figure 1.

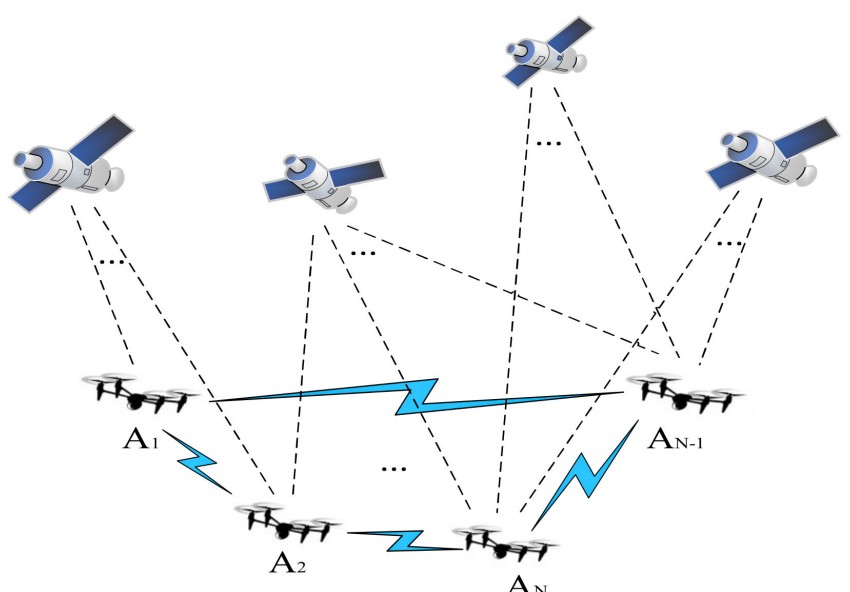

**Figure 1.** Cooperative positioning network of a UAV cluster.

In Figure 1, $A_1$, $A_2$, $A_{N-1}$, and $A_N$ all represent UAVs, while $A_N$ represents the n-th UAV. UAV positioning can be realized by a base station or cluster UAV. In the figure, the blue link represents the ranging communication link between UAVs. Meanwhile, the UAV is equipped with a satellite receiver and inertial measurement unit (IMU), which can receive satellite navigation signals and information such as speed and acceleration.

The essence of information geometry is to study the intrinsic geometric properties of probability distribution manifolds. The basic problems of probability theory and informa-

tion theory are geometrized by using the differential geometry method. Different types of probability distribution function families have corresponding statistical manifolds with certain structures. A statistical manifold can be understood as a surface in the parameter space of the probability density function. Every point on it corresponds to a specific probability distribution, and the coordinates of points are related to the parameters of the probability distribution. The form of each probability distribution function determines the relationship between its neighboring probability distribution function and the spatial structure it constitutes. The relationship between the probability distribution and statistical manifold is shown in Figure 2. In Figure 2, $X$ is the sample space of navigation source information, its probability density function is $p(x|\theta)$, $\theta$ is the parameter vector of $p(x|\theta)$, $\Theta$ is the vector space of parameter $\theta$, and $S$ is the statistical manifold with the parameter $\theta$ as the coordinate.

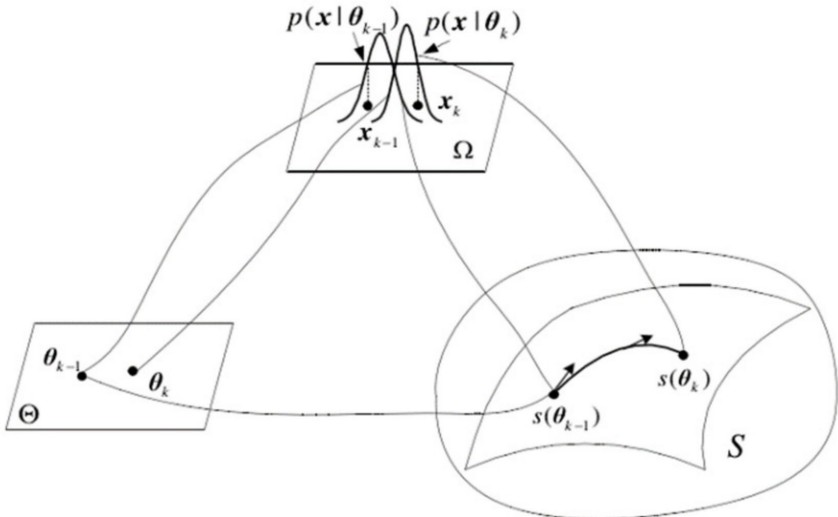

**Figure 2.** Schematic diagram of the statistical manifold.

In the multisource fusion scenario, combined with information geometry theory, the probability distribution functions of multiple UAV navigation information sources can be mapped to the Riemannian space as a family of functions to form a statistical manifold, and each point on the statistical manifold represents a probability distribution function. The multisource fusion algorithm based on information geometry theory can process heterogeneous data with better real-time performance, fault tolerance, and positioning accuracy.

The information fusion of a target UAV includes two parts: the target UAV's own information fusion and the cooperative information fusion from its neighboring UAVs. Each UAV merges the location information of its own multiple navigation sources, and then the target UAV iterates the information fusion through the location information provided by the neighbor UAVs and combines the ranging information to reduce the error. Figure 3 shows the multisource fusion process based on information geometry. Assume that UAV $A_1$ is the target UAV that needs to obtain positioning information. First, each information source of UAV $A_1$ is processed, a statistical manifold is established, the navigation information is mapped to the manifold, and the probability density function of each information source is obtained. Then, combined with the probability density function of location information provided by each neighboring UAV, the multisource fusion algorithm based on information geometry replaces the geodesic distance of the correlation matrix of the two probability density functions on the manifold by K–L divergence, obtains the lower bound of the fused objective function, and the real-time positioning result of target UAV $A_1$ is finally obtained.

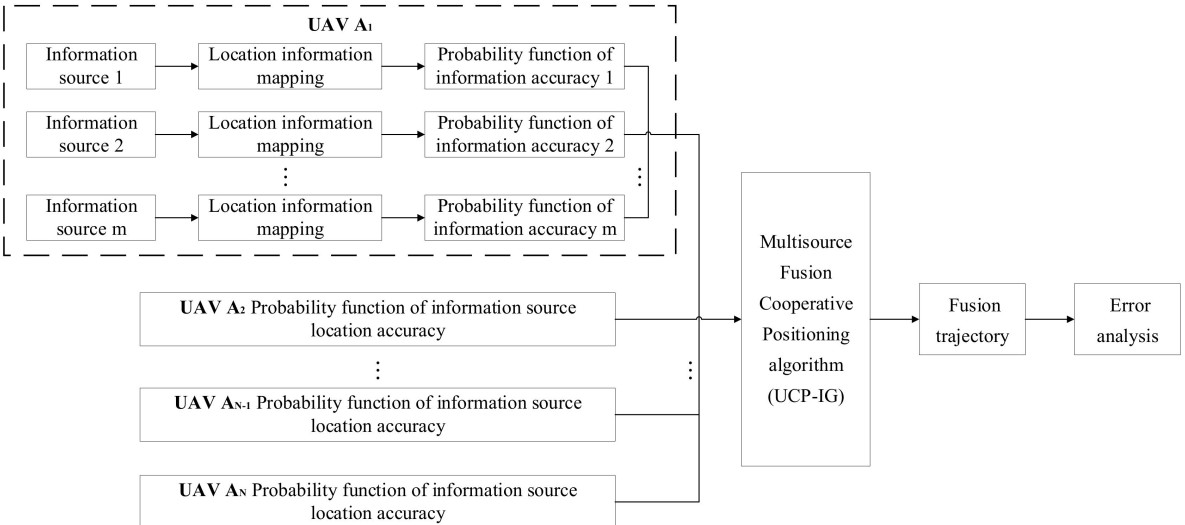

**Figure 3.** Flow chart of UAV cluster multisource fusion based on information geometry.

## 4. Multisource Fusion Cooperative Positioning Algorithm

Distributed navigation sources on UAV clusters contain $N$ navigation sources, whose observation data $x \in R^m$. At the $k$-th navigation source, the probability density of $x$ is $p_k(x|\theta), k = 1, \ldots, N$, and the correlation between navigation sources is unknown. The ultimate goal is to fuse all of these probability densities into a single probability density, denoted as $\hat{p}(x|\theta)$. For the convenience of the subsequent derivation process, $\hat{p}(x|\theta)$ is denoted as $\hat{p}$, and $p_k(x|\theta), k = 1, \ldots, N$ is denoted as $p_k, k = 1, \ldots, N$.

Assume that all $i$ in $k = 1, \ldots, N$, $p_k$ belong to the same probability distribution family, $S = \{p(x|\theta)\}$. Generally, an optimization criterion is needed to select the best point $\hat{p}$ on $S$. In the framework of information geometry, the distribution family $S$ is regarded as a Riemannian manifold, which is equipped with a Fisher information matrix [34]. Under this condition, the information-related geodesic distance can be obtained, and the information fusion theory criterion can be further obtained:

$$\hat{p} = \underset{p(x;\theta)\in S}{\operatorname{argmin}} \sum_{k=1}^{N} d^2(p_k, p(x|\theta)) \tag{1}$$

It is observed that the objective function in (1) is the sum of squared geodesic distances between a probability density and all locally known probability densities. However, as the information geometric analysis of statistical manifold is far from being fully studied, there is not much work to be done on (1). Even on many common statistical manifolds, a display expression for geodesic distance cannot be obtained. Therefore, attention needs to be focused on the family of Gaussian distributions. Although there is no definite form of geodesic distance between any two multivariate Gaussian distributions, the Gaussian family of distributions is one of the most studied manifold structures among various probability distribution families.

This paper assumes that the observed data $x$ probability density $p_k$ follows a Gaussian distribution, namely, $p_k(x|\theta) = N(\hat{x}_k, P_k)$, where $\hat{x} \in R^m$ and $P_k \in S_+^m$ are their respective mean and covariance matrices. Similarly, the probability density function after the fusion is defined as $\hat{p}(x|\theta) = N(\hat{x}, P)$. Under an assumption of a Gaussian distribution, the normal distribution with mean $\mu$ and variance $\Sigma$ is expressed as $N(\mu, \Sigma)$; then, the information fusion criterion (2) becomes:

$$(\hat{x}, R) = \underset{(\mu,\Sigma)\in R^m \times S_+^m}{\operatorname{argmin}} \sum_{k=1}^{N} d^2(p_k, N(\mu, \Sigma)) \tag{2}$$

Due to the lack of an exact expression of the objective function (2), directly solving the optimization problem is too complicated for the real-time requirements of UAV cluster positioning. From the perspective of information geometry, (2) has been transformed into a geodesic distance problem on a statistical manifold. However, the length of the geodesic is not easy to calculate, so divergence can be introduced as a distance function to measure the difference between two points on a manifold, where Kullback–Leibler divergence (KLD) is often used to measure the difference between two points on a statistical manifold [35], providing a lower limit for the minimum of the objective function in (2). The minimum value was solved by Kullback–Leibler minimization (KLM) to achieve UAV cluster fusion location optimization. KLD is defined as:

$$d_{KLD}(\theta_0, \theta_1) = E_{\theta_0}[\ln p(x|\theta_0) - \ln p(x|\theta_1)] \tag{3}$$

where $E[\cdot]$ represents the expectation according to the distribution $p(x|\theta)$. KLD has information monotony; that is, the divergence value will increase with the increase in the information difference between the two matrices, and vice versa. KLD can therefore be thought of as a distance on a manifold. However, it should be pointed out that the length of geodesic represents the shortest distance between two points on a manifold, and has good properties of distance measure, that is, symmetry, non-negativity, and triangle inequality. KLD does not satisfy symmetry and triangle inequality. Therefore, average KLD can be considered to satisfy the symmetry of distance measurement:

$$d_{KLD\_ave}(\theta_0, \theta_1) = \frac{1}{2}(d_{KLD}(\theta_0, \theta_1) + d_{KLD}(\theta_1, \theta_0)) \tag{4}$$

Consider that $x$ follows a complex Gaussian vector distribution, $N(0, R)$, whose mean is 0 and correlation matrix is $R$, and the expression is:

$$p(x) = \frac{1}{\pi^n |R|} \exp\left(x^H R^{-1} x\right) \tag{5}$$

where $|\cdot|$ represents the determinant of the matrix. Consider the probability distribution family $S = \{p(x|R)|R \in H(n)\}$ as parameterized by the correlation matrix $R \in H(n)$, where $H(n)$ is the $n \times n$-dimensional Hermitian positive definite matrix space. According to the theory of information geometry, $S$ can form a manifold with $R$ as its natural coordinate under certain topological and differential structures. Because coordinate $R$ of manifold $S$ is a correlation matrix, it can also be called a matrix manifold.

The zero-mean Gaussian vector distribution belongs to the exponential distribution family, which has a dual structure; that is, the manifold has two coordinate systems that are mutually dual and can be transformed into each other by the Legendre transformation of the potential function. The exponential distribution family has the following form [36]:

$$p(x|\theta) = \exp\left\{C(x) + \theta^T F(x) - \psi(\theta)\right\} \tag{6}$$

where $C(x)$ is a polynomial about $x$, $F(x)$ is a sufficient statistic of natural parameter $\theta$, $\psi(\theta)$ is called the potential function of the distribution, and the potential function of the zero-mean complex Gaussian distribution, $N(0, R)$, is [37]:

$$\psi(R) = -\log(|R|) \tag{7}$$

Suppose that the dual coordinate system of the natural coordinate $R$ is $R^\omega$, and the potential function of the manifold $S$ in the dual coordinate is $\phi(R^\omega)$; then, the natural coordinate $R$ and the dual coordinate $R^\omega$ have the following Legendre transformation relationship [37]:

$$\begin{cases} R = \nabla\phi(R^\omega) = \nabla\phi(\nabla\psi(R)) \\ R^\omega = \nabla\psi(R) \end{cases} \tag{8}$$

where $\nabla$ represents the gradient operator. Let $R_1$ and $R_2$ be any two points on the manifold $S$, and their dual coordinates are $(R_1, R_1^\omega)$ and $(R_2, R_2^\omega)$, respectively. Then, the Bregman divergence from $R_1$ and $R_2$ is defined as [37]:

$$d(R_1, R_2) = \psi(R_1) + \phi(R_2^\omega) - R_1 \cdot R_2^\omega \tag{9}$$

Substituting (8) into (9), the Bregman divergence can be expressed only by natural coordinates as:

$$d(R_1, R_2) = \psi(R_1) + \psi(R_2) - \nabla\psi(R_2)(R_1 - R_2) \tag{10}$$

Then, according to (7), substituting the potential function of the zero-mean complex Gaussian distribution, the Kullback–Leibler divergence (KLD) from $R_1$ to $R_2$ on the manifold is [38]:

$$d_{KLD}(R_1, R_2) = \mathrm{tr}\left(R_2^{-1}R_1 - I\right) - \log\left(\left|R_2^{-1}R_1\right|\right) \tag{11}$$

where $tr(\cdot)$ represents the trace of the matrix.

For $N$ Gaussian distributions $N(\mu_k, R_k), k = 1, \ldots, N$, if they are merged into a probability density, the problem is reduced to finding $N(\mu, R)$, so that the objective function obtains the minimum value:

$$f(u, R) = \frac{1}{N}\sum_{k=1}^{N} D^2(N(\mu_k, R_k), N(u, R)) \tag{12}$$

Here, the distance $D(\cdot)$ is the geodesic distance between the probability densities. However, there is no direct, explicit expression for this distance, so the optimization result of (12) cannot be obtained directly. K–L divergence is used instead in this paper, and so (12) is approximated by dividing it into two steps to solve the minimization problem of the objective function.

The first step: $N$ covariance matrices $R_k$ and $k = 1, \ldots, N$ of the Gaussian distribution are fused, so the problem becomes one of obtaining an R that satisfies the minimum value of the objective function (13):

$$f(R) = \sum_{k=1}^{N} d_{KLD}{}^2(R, R_k) \tag{13}$$

The second step: After determining the minimum covariance matrix $R$ of the objective function, the fusion result of the mean value $\hat{x}$ can be obtained through (14):

$$\hat{x} = R\sum_{k=1}^{N} \omega_k R_k^{-1} x_k \tag{14}$$

where $\omega_k$ is the weighting factor. Calculating the optimal weight factor $\omega_k$ requires repeated iterations, and the amount of calculation is very large. To avoid such a large amount of calculation, this algorithm uses the noniterative fast covariance intersection (FCI). The following discusses the value of the fusion weighting factor when $N = 2$ and $N \geq 2$.

When $N = 2$, the noniterative fusion weights $\omega_1$ and $\omega_2$ must satisfy $\omega_1 + \omega_2 = 1$. The linear constraint can be expressed as:

$$tr(R_1)\omega_1 - tr(R_2)\omega_2 = 0 \tag{15}$$

Thus, we obtain (16):

$$\begin{bmatrix} tr(R_1) & tr(R_2) \\ 1 & 1 \end{bmatrix}\begin{bmatrix} \omega_1 \\ \omega_2 \end{bmatrix} = \begin{bmatrix} 0 \\ 1 \end{bmatrix} \tag{16}$$

As long as $tr(R_1) + tr(R_2) > 0$ is established, the unique solution of (16) can be obtained:

$$\begin{aligned} \omega_1 &= \frac{tr(R_2)}{tr(R_1)+tr(R_2)} \\ \omega_2 &= \frac{tr(R_1)}{tr(R_1)+tr(R_2)} \end{aligned} \tag{17}$$

When $N \geq 2$, the noniterative fusion weight $\omega_1, \omega_2, \ldots, \omega_N$ satisfies $\omega_1 + \omega_2 + \ldots + \omega_N = 1$, and the linear constraint can be generalized to:

$$tr(R_k)\omega_k - tr(R_{k+1})\omega_{k+1} = 0, k = 1, 2, \ldots, N-1 \tag{18}$$

Let $\varepsilon_k = tr(R_k)$; then, (19) can be obtained:

$$\begin{bmatrix} \varepsilon_1 & -\varepsilon_2 & 0 & \cdots & 0 \\ 0 & \varepsilon_2 & -\varepsilon_3 & \cdots & 0 \\ \cdots & \cdots & \cdots & \cdots & \cdots \\ 0 & \cdots & 0 & \varepsilon_{N-1} & -\varepsilon_N \\ 1 & \cdots & 1 & 1 & 1 \end{bmatrix} \begin{bmatrix} \omega_1 \\ \omega_2 \\ \cdots \\ \omega_{N-1} \\ \omega_N \end{bmatrix} = \begin{bmatrix} 0 \\ 0 \\ \cdots \\ 0 \\ 1 \end{bmatrix} \tag{19}$$

The analytical expression of the fusion weights $\omega_1, \omega_2, \ldots \omega_N$ can be derived from (19), as is shown in (20):

$$\omega_k = \frac{\prod\limits_{i=1, j \neq k}^{N} \varepsilon_i}{\sum\limits_{i=1}^{N} \prod\limits_{j=1, j \neq i}^{N} \varepsilon_j} = \frac{\frac{1}{\varepsilon_k}}{\sum\limits_{i=1}^{N} \frac{1}{\varepsilon_k}} \tag{20}$$

To obtain the solution to $R$ from (13), the gradient descent method is used to iteratively solve the solution of the objective function, where the initial iterative value of $R$ is:

$$R = \left( \sum_{k=1}^{N} \omega_k R_k^{-1} \right)^{-1} \tag{21}$$

By the matrix derivative formula and the matrix derivative chain rule:

$$\frac{\partial \ln|A|}{\partial A} = \left( A^{-1} \right)^T \tag{22}$$

$$\frac{\partial(AB)}{\partial A} = \frac{\partial(BA)}{\partial A} = B^T \tag{23}$$

$$\frac{\partial z}{\partial X} = \left( \frac{\partial Y}{\partial X} \right)^T \frac{\partial z}{\partial Y} \tag{24}$$

Taking the derivative of (13), we can obtain:

$$f'(R) = \sum_{k=1}^{N} 2d_{KLD}(R, R_k) \left( R_k^{-1} - R^{-1} \right) \tag{25}$$

Therefore:

$$R_{i+1} = R_i - hf'(R_i) \tag{26}$$

$h$ is the iteration step size.

The entire process of the UCP-IG method based on matrix K–L divergence square mean minimization is shown in Algorithm 1.

---

**Algorithm 1:** UCP-IG Fusion Algorithm

---

**Input:** $\left\{\left(\hat{x}_k, R_k\right)\right\}_{k=1,\ldots,N}$, step size $0 \leq h \leq 1$

**Output:** $\left(\hat{x}, R\right)$

1: Initialize $R$ using equation (21);
2: **while** *not converged* **do**
3:  |  Run iteration: $R \leftarrow R - hf'(R)$;
4: **end**
5: Compute $\omega_k, k = 1,\ldots,N$ using equation (20);
6: Calculate $\hat{x}$ using equation (14);
7: **Return** $\hat{x}, R$.

---

## 5. Simulation Results and Analysis

In this section, we take four mobile UAVs as examples and apply the multisource fusion cooperative positioning algorithm to the cooperative positioning of UAV clusters in different experimental environments. The experiment consists of three scenes. The first scene is a comprehensive scene in which each UAV flies freely. In the second scenario, the UAV cluster flies randomly in the experimental site and sends back experimental data in real time through the GUI interface. The third scene is the occlusion scene. In the moving process of the UAV cluster, some UAV navigation source signals will be lost in flight due to occlusion by building groups.

### 5.1. Experimental Environment

The experimental site selects some buildings of Xi'an's classic landmarks, the Big Wild Goose Pagoda and the Da Ci'en Temple, and builds them under the guidance of actual aerial photos, as shown in Figure 4a. In this paper, the aerial photos of the scenic spot are scaled down and painted, and the painting specifications are long and wide. The landmark buildings in the figure are simulated by using cardboard boxes and placed in corresponding positions, respectively. The established test positioning site is shown in Figure 4b.

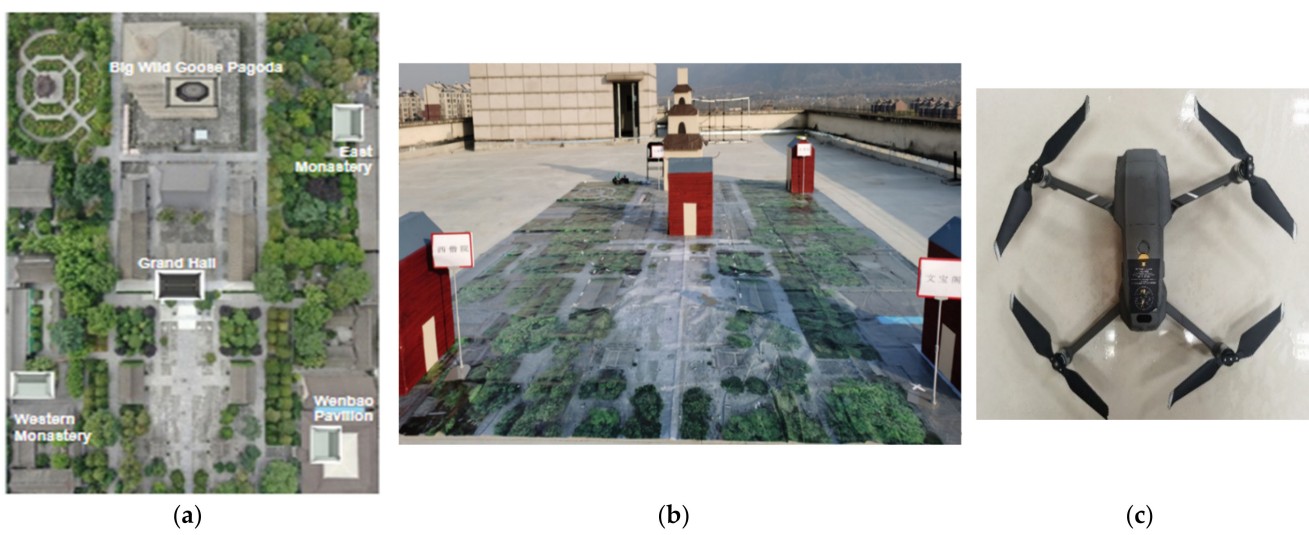

(a)            (b)            (c)

**Figure 4.** (**a**) Aerial photo of the local pagoda. (**b**) Experimental environment. (**c**) UAV used in experiments.

Figure 4a contains five iconic attractions, namely the Big Wild Goose Pagoda, West Monastery, East Monastery, Wenbao Pavilion, and Grand Hall. The drone uses four DJI Mavic2 aerial photography quadrotors and the model is zoom type, as shown in Figure 4c.

### 5.2. Comprehensive Scenario

The experimental conditions are as follows: there are four UAVs throughout the measured experimental environment, which are equipped with satellite navigation receivers, IMUs, and communication equipment. The measured noise is considered Gaussian noise, and all UAVs can distance and communicate with other UAVs. A commercial IMU inertial measurement module, model LPMS-IG1, was used in the simulation. The zero stability is 4 degrees/h, the resolution is 0.01 degrees, and the sampling rate is 100 Hz. Table 1 lists the remaining simulation parameters.

**Table 1.** Parameter settings.

| Parameter | Value |
| --- | --- |
| Number of UAVs | 4 |
| Time | 20 s |
| Number of Monte Carlo simulations | 10,000 |
| Drone motion jitter error | 0.5 m |
| Satellite navigation ranging information error | 1.5 m |

Based on the above simulation conditions, the UCP-IG method is compared with the LS method proposed in [25], the UKF method proposed in [22], and the neural network method proposed in [28] in comprehensive scenarios. The initial positions of each UAV in the UAV group were: UAV $A_1$ (0.97 m, 0.02 m, 0.22 m), UAV $A_2$ (4.29 m, 4.76 m, 0.26 m), UAV $A_3$ (0.22 m, 4.77 m, 0.27 m), and UAV $A_4$ (3.14 m, 0.49 m, 0.29 m). Each UAV makes random motions from the initial position, and the specific motion trajectories of the four UAVs are shown in Figure 5. After 10,000 Monte Carlo simulations, the UAV motion positioning results of four algorithms are shown in Figure 6, where the purple motion trajectory represents the LS method positioning result. The green motion trajectory represents the UKF method positioning result, and the blue motion trajectory represents the ANN method. The red motion track represents the positioning result of the UCP-IG method. The corresponding positioning error fluctuations of the four algorithms are shown in the GUI error display platform in Figure 7.

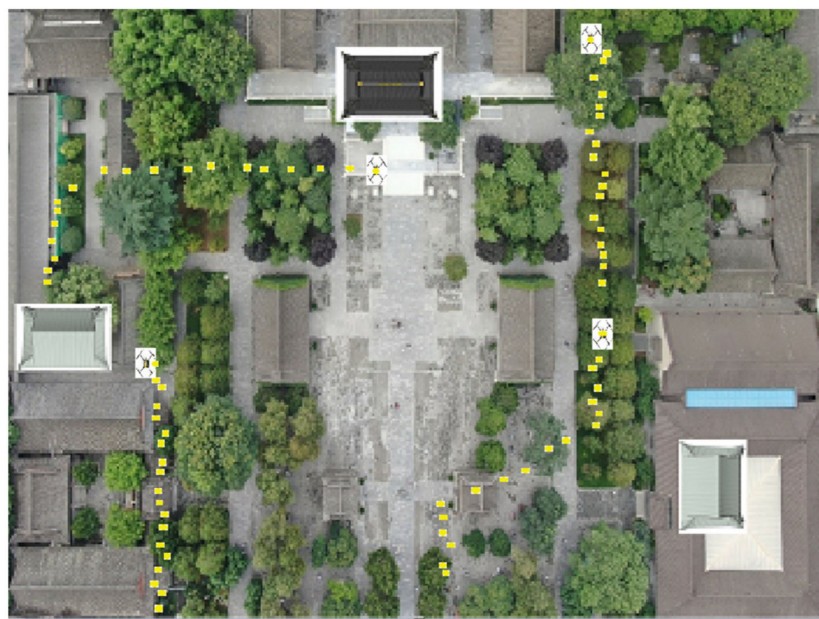

**Figure 5.** UAV trajectory (comprehensive scenario).

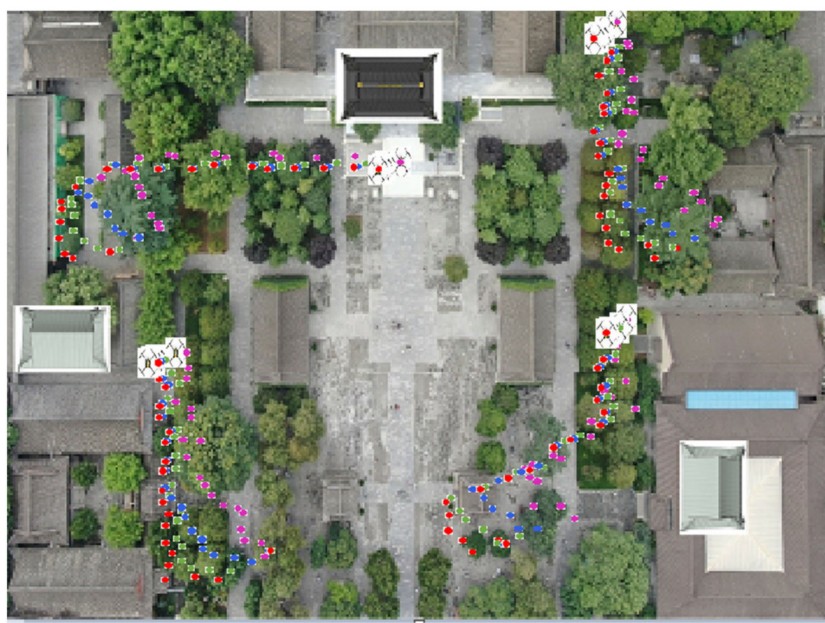

**Figure 6.** LS, UKF, ANN, UCP-IG method positioning trajectory (comprehensive scenario).

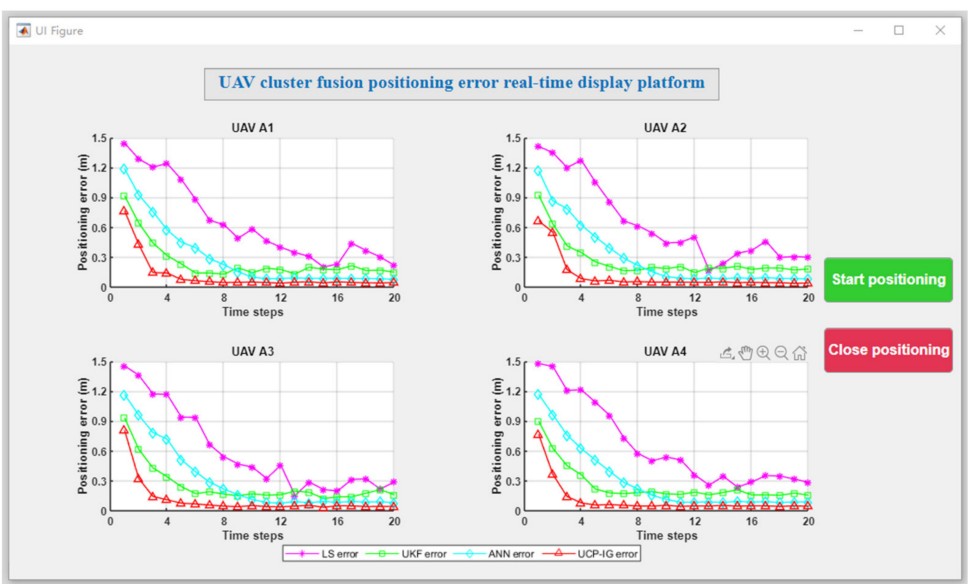

**Figure 7.** GUI positioning error display platform (comprehensive scenario).

In Figure 6, various UAV swarm positioning methods can achieve fusion and ensure that the UAV swarm positioning is in a stable state. In order to further compare the positioning error, the fusion positioning error of each UAV at each moment is compared with the actual position, and the results are shown in Figure 7. It can be seen from Figure 7 that the error of the LS method after the convergence is between 0.21 and 0.43 m, and the error jitter is obvious; the error fluctuation of the UKF method after convergence is between 0.13 and 0.18 m, and the error jitter is higher than that of the ANN method. The error fluctuation of the ANN method is about 0.07 m, and the error jitter is small. It shows that the multi-layer neurons of the ANN method can effectively suppress the error jitter of the UAV cluster. The UCP-IG method proposed in this paper achieves the shortest convergence time, and the error jitter range is about 0.05 m, which is better than the single navigation source and other UAV positioning methods. It shows that the UCP-IG method can effectively reduce the jitter error of each navigation source by converting all kinds of

navigation source information into information geometric probability for fusion. At the same time, the positioning accuracy of the UAV cluster is improved. Table 2 lists the real height data and UCP-IG positioning height data of each UAV in this scenario. It can be seen that the data are very close, and the height error fluctuates at the centimeter level.

**Table 2.** Part of the experimental data (comprehensive scenario).

| UAV Number | | Time Steps | | | |
|---|---|---|---|---|---|
| | | **5** | **10** | **15** | **20** |
| UAV $A_1$ | True height (m) | 1.05 | 2.05 | 3.04 | 4.06 |
| | Location height of UCP-IG (m) | 1.09 | 2.08 | 3.06 | 4.08 |
| UAV $A_2$ | True height (m) | 1.10 | 2.07 | 3.03 | 4.07 |
| | Location height of UCP-IG (m) | 1.14 | 2.11 | 3.05 | 4.09 |
| UAV $A_3$ | True height (m) | 1.01 | 2.04 | 3.05 | 4.05 |
| | Location height of UCP-IG (m) | 1.05 | 2.07 | 3.08 | 4.08 |
| UAV $A_4$ | True height (m) | 1.03 | 2.05 | 3.09 | 4.08 |
| | Location height of UCP-IG (m) | 1.06 | 2.08 | 3.11 | 4.11 |

In order to quantitatively analyze the positioning errors of UAV $A_1$, UAV $A_2$, UAV $A_3$, and UAV $A_4$, the average errors are statistically analyzed, and the results are shown in Table 3. It can be seen from the results in Table 3 that the average error of the LS method, UKF method, and ANN method is greater than that of the UCP-IG method, indicating that the UCP-IG method can effectively improve the stability of UAV clusters.

**Table 3.** Average positioning errors (comprehensive scenario).

| UAV Number | Average Error (m) | | | |
|---|---|---|---|---|
| | **LS** | **UKF** | **ANN** | **UCP-IG** |
| UAV $A_1$ | 0.63 | 0.26 | 0.29 | 0.11 |
| UAV $A_2$ | 0.62 | 0.26 | 0.30 | 0.10 |
| UAV $A_3$ | 0.60 | 0.26 | 0.30 | 0.17 |
| UAV $A_4$ | 0.62 | 0.25 | 0.30 | 0.10 |

*5.3. Experimental Random Scenario*

In order to make the experimental results universal, the verification was carried out in the actual experimental scene. The initial positions of each UAV of the UAV group were: UAV $A_1$ (0.36 m, 0.46 m, 0.30 m), UAV $A_2$ (3.16 m, 5.28 m, 0.35 m), UAV $A_3$ (0.62 m, 4.04 m, 0.34 m), and UAV $A_4$ (2.22 m, 0.49 m, 0.32 m). Each UAV makes random motions from the initial position, and the specific motion trajectories of the four UAVs are shown in Figure 8. The LS, UKF, ANN, and UCP-IG methods are compared respectively, and the UCP-IG motion localization results of the four algorithms are shown in Figure 9, where the purple motion trajectory represents the LS method's localization result, the green motion trajectory represents the UKF method's localization result, and the blue motion trajectory represents the UKF method's localization result. The color motion trajectories represent the localization results of the ANN method, and the red motion trajectories represent the localization results of the UCP-IG method. The corresponding positioning error fluctuations of the four algorithms are shown in the GUI error display platform in Figure 10.

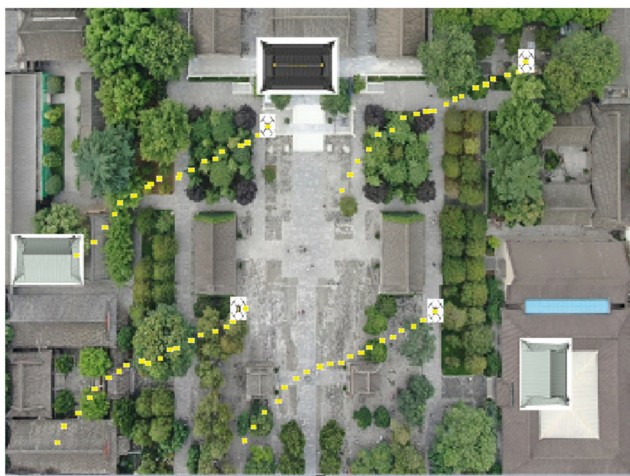

**Figure 8.** UAV trajectory (experimental random scenario).

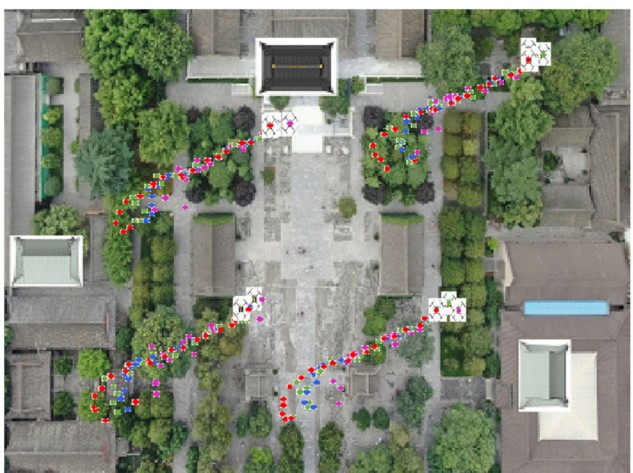

**Figure 9.** LS, UKF, ANN, UCP-IG method positioning trajectory (experimental random scenario).

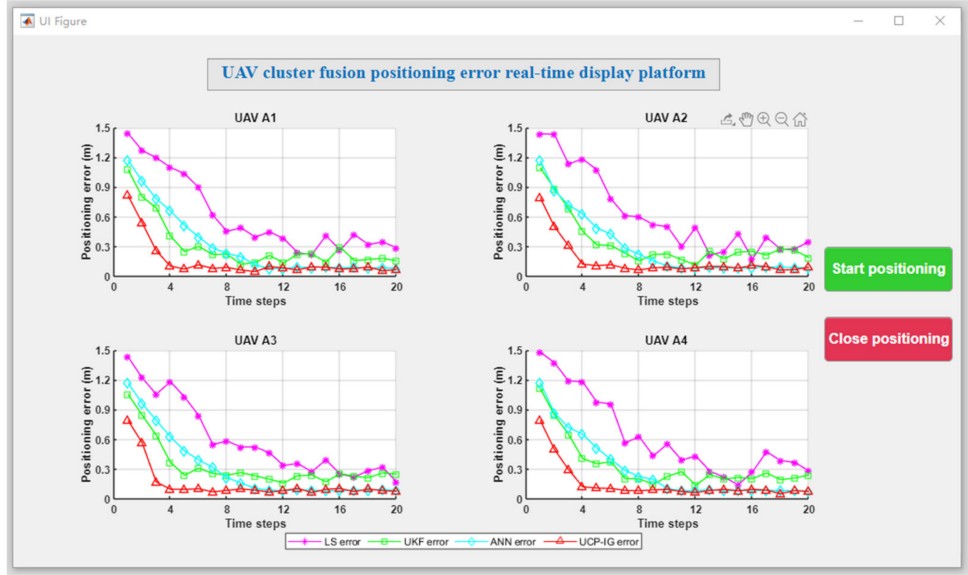

**Figure 10.** GUI positioning error display platform (experimental random scenario).

From Figures 9 and 10, it can be seen that the LS method, UKF method, ANN method, and UCP-IG method can all achieve convergence, where the LS method has a slight increase in the average value of the localization error fluctuation in random scenes, which is due to random motion In the LS method, when the direction and position are changed, the position of the previous moment has an impact on the next moment; the UKF method, ANN method and UCP-IG method have the average value of the fluctuation of the positioning error in the random experimental scene and the basic value in the simulation scene. Similarly, the positioning error of the UCP-IG method fluctuates around 0.08 m.

For the convergence time of gradient descent, the proposed cooperative localization method essentially realizes the interaction of localization information with the surrounding nodes. The information of the surrounding nodes is used to locate the whole UAV cluster. The gradient descent method brings some computational delay, but the number of cluster nodes involved in the location of a single UAV is limited. Therefore, the impact on the overall computational complexity delay is small, which can also be seen in Figure 10.

Table 4 shows the real height data and UCP-IG positioning height data of each UAV in the random motion experiment scenario, and the height error fluctuation is also at the centimeter level. In order to further compare the performance of each algorithm, the average error of each algorithm is compared, as shown in Table 5.

**Table 4.** Part of the experimental data (experimental random scenario).

| UAV Number | | Time Steps | | | |
|---|---|---|---|---|---|
| | | 5 | 10 | 15 | 20 |
| UAV $A_1$ | True height (m) | 1.57 | 3.05 | 4.56 | 6.07 |
| | Location height of UCP-IG (m) | 1.61 | 3.10 | 4.60 | 6.11 |
| UAV $A_2$ | True height (m) | 1.55 | 3.04 | 4.55 | 6.02 |
| | Location height of UCP-IG (m) | 1.59 | 3.10 | 4.57 | 6.05 |
| UAV $A_3$ | True height (m) | 1.59 | 3.00 | 4.52 | 6.05 |
| | Location height of UCP-IG (m) | 1.64 | 3.03 | 4.56 | 6.09 |
| UAV $A_4$ | True height (m) | 1.51 | 3.01 | 4.55 | 6.04 |
| | Location height of UCP-IG (m) | 1.55 | 3.04 | 4.59 | 6.07 |

**Table 5.** Average positioning errors (experimental random scenario).

| UAV Number | Average Error (m) | | | |
|---|---|---|---|---|
| | LS | UKF | ANN | UCP-IG |
| UAV $A_1$ | 0.62 | 0.33 | 0.29 | 0.14 |
| UAV $A_2$ | 0.60 | 0.32 | 0.29 | 0.13 |
| UAV $A_3$ | 0.63 | 0.35 | 0.30 | 0.15 |
| UAV $A_4$ | 0.65 | 0.33 | 0.29 | 0.15 |

As can be seen from Table 5, the average error of the LS method fluctuates between 0.60 and 0.65 m in the measured scene. The average error of the UKF method fluctuates around 0.33 m. The ANN method is better and fluctuates about 0.29 m. The UCP-IG method is the best and fluctuates around 0.14 m. The UCP-IG method proposed in this paper can transform all types of navigation sources into information probability models, which is beneficial to the suppression of navigation information errors. The corresponding KL fusion method can realize the fast fusion of the distributed navigation source information of UAV clusters.

## 5.4. Experimental Occlusion Scenario

In the process of UAV movement, satellite navigation sources may lose signal or offset error due to the occlusion of buildings and environmental interference. In this section, the initial positions of the UAVs were set as follows: UAV $A_1$ (1.17 m, 0.98 m, 0.22 m), UAV $A_2$ (4.29 m, 4.75 m, 0.26 m), UAV $A_3$ (0.42 m, 6.56 m, 0.27 m), and UAV $A_4$ (2.24 m, 0.50 m, 0.30 m). The movement trajectory of the four UAVs is shown in Figure 11. In this occlusion scenario, satellite signal loss or navigation signal deviation occurs due to occlusion. Among them, UAV $A_1$ and UAV $A_3$ lost satellite signal in unit time 6–10, and UAV $A_4$ had navigation signal deviation in unit time 11–16. The motion positioning results of the UAVs with four algorithms are shown in Figure 12, in which the purple trajectory represents the positioning results of the LS method, the green trajectory represents the positioning results of the UKF method, the blue trajectory represents the positioning results of the ANN method, and the red trajectory represents the positioning results of the UCP-IG method. The corresponding positioning error fluctuation of the four algorithms is shown in Figure 13 on the GUI error display platform.

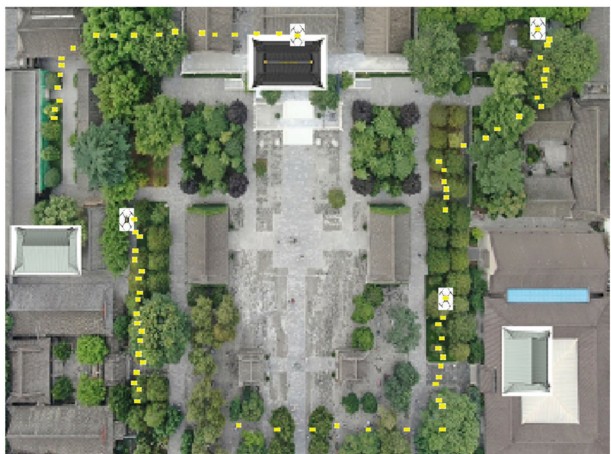

**Figure 11.** UAV trajectory (experimental occlusion scenario).

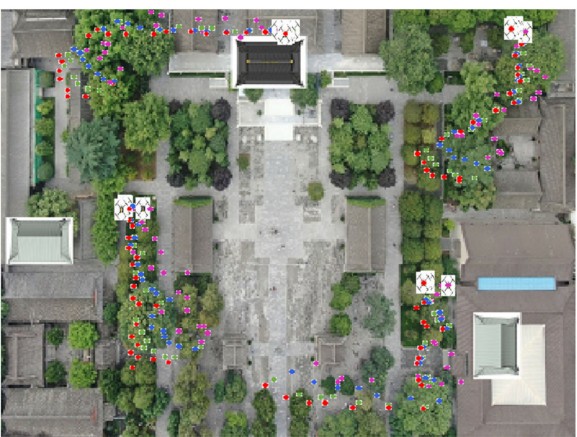

**Figure 12.** LS, UKF, ANN, UCP-IG method positioning trajectory (experimental occlusion scenario).

It can be seen from Figures 12 and 13 that the satellite signal loss of UAV $A_1$ and UAV $A_3$ occurs at the eighth time unit, and the LS method can recover the UAV positioning error by about 0.45 m after six time units. It is because LS suppresses the error caused by the lost signal by means of attenuation. The UKF method can recover the UAV positioning error to about 0.19 m after four time units. The ANN method can restore the UAV positioning error to the normal range of 0.08 m after five time units, indicating that although the ANN method can well integrate multi-type navigation source information, when the navigation

source information is lost, it cannot quickly eliminate the impact of the loss. The UCP-IG method proposed in this paper can restore the UCP-IG positioning error to about 0.07 m within two time units, which is superior to other UCP positioning methods and single navigation source accuracy. When the navigation signal of UAV $A_4$ deviates at the 12th time unit, the time positioning error of LS is restored to about 0.48 m after six time units, and the positioning error of UKF is restored to about 0.19 m after four time units. The time positioning error of five time units is restored to about 0.08 m, and the time positioning error of the UCP-IG method is restored to about 0.07 m after two time units. There are various types of navigation sources on the man–machine, and the fusion speed is the fastest, which can effectively suppress the influence of the loss of navigation sources and interference errors. In addition, Table 6 lists the real height data and UCP-IG positioning height data of each UAV in the random motion experiment scenario. Even if there is signal occlusion, the height error fluctuation after fusion remains at the centimeter level.

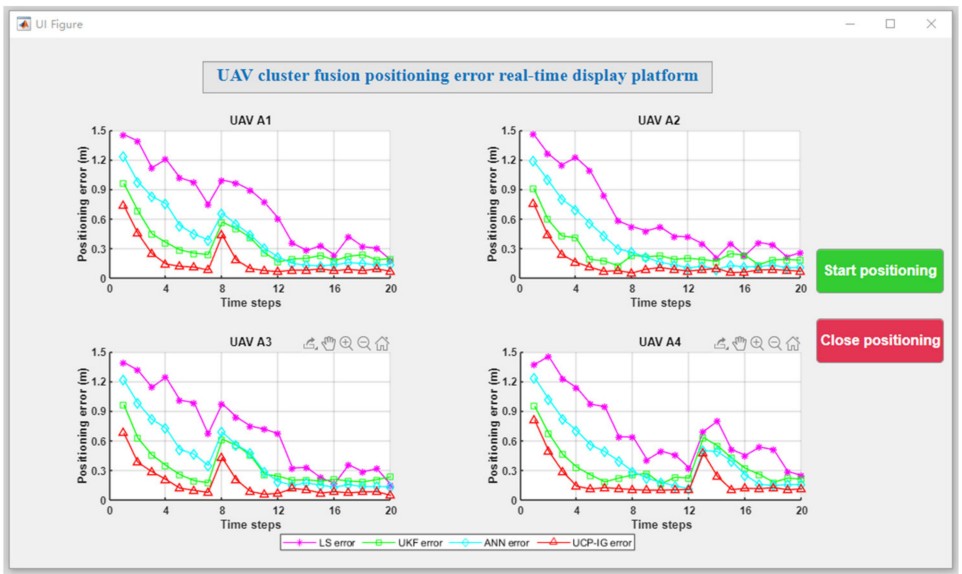

**Figure 13.** GUI positioning error display platform (experimental occlusion scenario).

**Table 6.** Part of the experimental data (experimental random scenario).

| UAV Number | | Time Steps | | | |
| --- | --- | --- | --- | --- | --- |
| | | 5 | 10 | 15 | 20 |
| UAV $A_1$ | True height (m) | 1.05 | 2.06 | 3.04 | 4.06 |
| | Location height of UCP-IG (m) | 1.14 | 2.13 | 3.09 | 4.14 |
| UAV $A_2$ | True height (m) | 1.10 | 2.07 | 3.04 | 4.07 |
| | Location height of UCP-IG (m) | 1.15 | 2.12 | 3.09 | 4.10 |
| UAV $A_3$ | True height (m) | 1.01 | 2.04 | 3.05 | 4.05 |
| | Location height of UCP-IG (m) | 1.10 | 2.10 | 3.09 | 4.12 |
| UAV $A_4$ | True height (m) | 1.03 | 2.05 | 3.09 | 4.08 |
| | Location height of UCP-IG (m) | 1.07 | 2.09 | 3.14 | 4.13 |

## 6. Conclusions

Aiming to address the problems of UAV cluster jittering, the small number of navigation sources, and the high real-time demand, this paper proposes a multisource fusion UAV cluster cooperative positioning algorithm. The method creatively uses the correlation between UAV navigation source information probability and positioning accuracy and

converts different types of navigation source information into probability functions to realize fast fusion. To analyze the results achieved by the UCP-IG method proposed in this paper, a comparison was performed with the current mainstream UAV cluster positioning technologies; namely, the proposed method was compared with the LS and ANN methods in ideal, occlusion, and random scenarios. The results show that the UCP-IG method can make full use of the distributed navigation source information of different types carried by the UAV cluster to realize fusion and effectively improve the positioning accuracy and stability of UAV clusters.

**Author Contributions:** Conceptualization, C.T. and Y.W.; methodology, C.T.; software, Y.W.; validation, C.T., L.Z. and Y.Z.; formal analysis, L.Z.; investigation, C.T.; resources, L.Z.; data curation, Y.W.; writing—original draft preparation, Y.W.; writing—review and editing, L.Z.; visualization, L.Z.; supervision, H.S.; project administration, H.S.; funding acquisition, C.T. and L.Z. All authors have read and agreed to the published version of the manuscript.

**Funding:** This research was funded by National Natural Science Foundation of China under Grant 62171735, 62271397, 62101458 and 61803310, in part by the Natural Science Basic Research Program of Shaanxi under Grant 2021GY-075, and in part by China Postdoctoral Science Foundation under Grant 2020M673485.

**Institutional Review Board Statement:** Not applicable.

**Informed Consent Statement:** Not applicable.

**Conflicts of Interest:** The authors declare no conflict of interest.

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
