# Peer review of "Multisource Fusion UAV Cluster Cooperative Positioning Using Information Geometry"

_remotesensing, doi:10.3390/rs14215491_

Round 1

Reviewer 1 Report

The authors present an information fusion model for better positioning of a UAV team using information geometry. The authors use up to 4 UAVs to test the proposed technique. 

The paper needs moderate language editing. There are a few sentence structure errors and a few typos in the paper. For example, LS is not defined before its use in the abstract. T in the word "time" in Figure 8 should be capitalized. The last but one sentence in pp. 14 is too long -- it goes on from line 440 to 448. Please ensure that you break such long sentences throughout the paper into smaller ones for better readability. Furthermore, I believe that the title of the paper itself can be phrased better. Please look into it.

Regarding the experiments, why did you choose only 4 robots? What will happen if you have only 2 or 10, for example?

Fig. 18 presents the average error results, which clearly show the advantage of using the proposed technique. However, if we look at Fig. 16, we can see that ANN performs similarly to UCP-IG at the end. Therefore, using only the average values to judge the ANN-based technique might be misleading. Instead, I suggest using the last error value as the performance metric potentially along with the average error values that you already have. 

Figs. 5, 6, and 7 can be placed side-by-side to make the paper more compact.

Author Response

Thank you for your letter and for the reviewers’ comments concerning our manuscript entitled “Multi-source Fusion UAV Cluster Cooperative Positioning Using Information Geometry”. These comments were valuable and helpful in revising and improving our paper. We studied the comments carefully and have made corrections that we hope will meet with your approval. The responses to the reviewers’ comments in attechment.

Reviewer 2 Report

Authors propose a new technique for cooperative localization of UAV clusters, which uses information geometry. Localization of UAV, specially in environments where the GPS signal can experiment problems, is today a problem to solve.

The paper is well organized. The background is solid, and I think that the method is clearly explained. Results demonstrate that the proposal improves the positioning accuracy of state-of-the art techniques (LS, UKF, and ANN).

Regarding the submitted manuscript, I have several minor comments that should be addressed by the authors:

- The wording of the entire manuscript needs to be revised, as there are several punctuation errors (some examples can be found in lines 114, 385, 424, 444, 448...).

- I would change from "Figure" to "Table" all the tables in the text.

- The figures showing the results should be shown in a larger size and separately (instead of directly using screenshots of the developed GUI). This would facilitate readability. 

In relation to the experiments, it seems that this is a simulation study, but authors refer to a real UAV model, equipped with GPS, IMU... It is not entirely clear to me which part was simulated and which part is "real". For example, do UAVs actually fly, and are the algorithms run in matlab? Do the UAV fly and also execute the algorithms, and Matlab is used to verify their behavoir? Are the UAV capable of measuring distances and communicate or this is simulated? To sum up, I have many doubts about the manner in which the experiments are conducted.

On the other hand, my main concern is that the use of the gradient descent method would involve a computational complexity that would make the proposal unfeasible in scenarios where real-time is a requirement that we cannot ignore. Communications cost, which is inherent to the cooperative behavior described, must also be considered when assessing the feasibility of the proposal for this type of scenarios. Authors should definitely discuss both points.

Author Response

Thank you for your letter and for the reviewers’ comments concerning our manuscript entitled “Multi-source Fusion UAV Cluster Cooperative Positioning Using Information Geometry”. These comments were valuable and helpful in revising and improving our paper. We studied the comments carefully and have made corrections that we hope will meet with your approval. The responses to the reviewers’ comments are in attachment.

Reviewer 3 Report

This paper describes a new methods for a group of UAVs to localise themselves using a multi-source fusion cooperative positioning algorithm. The effectiveness of this new method in experimentally established, in simulation, and on real hardware, on different comparative tests. The paper is clearly written, the subject is sufficiently well introduced, results are clearly illustrated and conclusions are sound. I only have minor corretions to suggest:

1) In Section 2 (Related work) please replace "Refernce" and "Literature" with more appropriate style of writing, such as "In the reserach work discussed in [....]" or "in the study illustrated in [...]"

2) Line 444, change "time The shortest" in "the shortest time".

3) Line 448 there is a problem with the text which is not clear. Please rephrase.

4) I would also suggest to increase the font of the axis labels in Fig.11, 16, and 21. 

Author Response

(The authors gave the same response as above.)

Round 2

Reviewer 1 Report

The authors have carefully responded to my previous comments. The paper can be accepted in its present form, in my opinion. 

Author Response

Thank you for your letter. It is a great honor to have your recognition of this work. At the same time, thank you very much for your work and consideration for the publication of our paper. On behalf of my co-authors, we would like to express our heartfelt thanks to the editors and reviewers.

Reviewer 2 Report

Authors have satisfactorily addressed most of my comments.

However, a discussion about the computational impact of the employ of the gradient descend method, one of the main concerns I pointed out in my previous review, is is still pending.

Author Response

Thank you for your letter and for the reviewers’ comments concerning our manuscript entitled “Multi-source Fusion UAV Cluster Cooperative Positioning Using Information Geometry”. These comments were valuable and helpful in revising and improving our paper.

Thank you for your approval of other issues. Gradient descent will increase the computational complexity, but in the method proposed in this paper, each UAV mainly realizes information interaction and positioning with the surrounding UAV nodes. If the ranging range is too large, the computational complexity will be greatly increased, so it is necessary to control the communication ranging range of UAV. This part has been modified in this paper。
